# OpenReview forum: "Misaligned Roles, Misplaced Images: Structural Input Perturbations Expose Multimodal Alignment Blind Spots"
_ICLR.cc/2026/Conference — ICLR 2026 Poster_

### Official Review · Reviewer_gSyD · 2025-10-27

**Soundness:** 3
**Presentation:** 3
**Contribution:** 2
**Rating:** 2
**Confidence:** 4

**Summary:**

This paper introduces Role-Modality Attacks (RMA) — a new class of adversarial attacks that exploit structural vulnerabilities in multimodal language models (MMLMs). Specifically, it manipulates the role tokens (swapping user/assistant) and modality token positions (moving the image token around) in standard chat templates. Experiments are conducted using 3 MLLMs (qwen2-vl, llava-1.5, and phi-3.5-vision) on two benchmarks, Advbench and HarmBench, showing obvious gains.

**Strengths:**

1. The paper is clearly rewritten and easy to follow.

**Weaknesses:**

1. **Should RMA be treated as Adversarial Attack:** I'm somewhat concerned about whether the proposed RMA is proper to be treated as 'adversarial attack'. The proposed RMA mainly involves reformatting the prompt template (e.g., swapping role tokens or moving the image token around). As LLMs are known to be sensitive to prompts (especially prompt templates), the resulting performance drop of applying RMA is quite expected. I think it may be more appropriate to treat it as the LLMs' sensitivity/robustness to prompts instead of actual "adversarial attack", since altering the prompt template will degrade model performance on all tasks, which is not specific to adversarial attack on jailbreaking.
2. **Practical Usefulness:** I'm also concerned about the practical usefulness of RMA. In reality users will not be able to modify the chat template when using APIs.
3. **Similar prior works & missing reference:** There is a previous workshop paper that also studies how the chat template can help jailbreak an aligned MLLM [1]. The objective in [1] seems to be pretty similar to this paper, yet there is no reference to [1].
4. **Experiment on more recent MLLMs:** I think it might be worthwhile to experiment RMA on newer MLLMs such as qwen2.5/qwen3, gemma 3, internvl2.5/3/3.5 etc.. These models are RL-finetuned, which may have different behaviors from non-RLed models tested in the paper. (e.g. in Table1, Qwen2-VL already demonstrates better robustness against RMA, compared to earlier models like LLava-1.5 and phi3.5-v).

[1] ChatBug: A Common Vulnerability of Aligned LLMs Induced by Chat Templates. NeurIPS 2024 Workshop SafeGenAI

**Questions:**

Please see weakness.

---

> ### Author Response · Authors · 2025-11-20
> **Response to Reviewer 4 (gSyD)**
>
> We appreciate your positive comment on the paper’s clarity.
>
> ---
>
> **Q1) RMA as Adversarial Attack or LLMs' sensitivity/robustness?**
>
> Thank you for this thoughtful question. We agree that RMAs differ from traditional content-based jailbreaks and instead operate at the _structural_ level of the prompt template while preserving the semantics. Our use of the term _adversarial attack_ follows its broader definition in the safety and robustness literature, where any input that systematically and intentionally elicits harmful behavior, even when semantics remain unchanged, qualifies as adversarial \[1, 2\]. While it is true that LLMs are generally prompt-sensitive across tasks, describing RMAs solely as robustness artifacts would understate their nature. Unlike generic prompt sensitivity, which causes untargeted performance degradation, RMAs are **directional and targeted**, deliberately exposing a failure mode where harmless structural changes selectively bypass safety alignment.
>
> Based on your suggestion, we have added **Appendix I (Conceptual Framing of Role–Modality Attacks)**, clarifying this distinction.
>
> \[1\] Goyal et al. A Survey of Adversarial Defences and Robustness in NLP, ACM Computing Surveys
>
> \[2\] Tramèr et al. Fundamental Tradeoffs between Invariance and Sensitivity to Adversarial Perturbations, ICML 2020
>
> ---
>
> **Q2) Practicality of RMA**
>
> This is a great question. Please see the response under Common Question 2.
>
> ---
>
> **Q3) Missing reference?**
>
> We appreciate your comment. The ChatBug paper is indeed relevant, and we have already cited and discussed it in detail in our **Related Work section** (Starting from line 441). For brevity, we encourage you to refer to that section for the full discussion.
>
> In short, our work extends beyond _ChatBug_’s focus on control-token mismatches and inconsistencies (e.g., _BOT_, _EOT_) in text-only LLMs by revealing an expanded attack surface in multimodal models, driven by the mere inclusion of new modalities and the combinatorial complexity of token positions. We identify two novel attack strategies that are effective even when format rules are preserved. Modality Manipulation succeeds by repositioning image tokens without introducing syntactic mismatches. In addition, Role Confusion goes beyond superficial structural perturbations and exposes a deeper alignment asymmetry between user and assistant roles (as noted by Reviewer 3), leading to harmful downstream behaviors as demonstrated in our synthetic conversation generation study and an example of Training data (SFT/RL) extraction risks **(see Ethics Statement and Appendix D)**. Finally, our latent-space analyses provide causal insights into representational shifts relative to the refusal directions, offering a complementary perspective to prior empirical observations.
>
> We revised the related work section to ensure these connections and distinctions are made even more explicit.
>
> ---
>
> **Q4) Additional experiments across model families**
>
> This was a great suggestion to improve our paper. Please see the response under Common Question 1.

---

### Official Review · Reviewer_iN6X · 2025-10-27

**Soundness:** 4
**Presentation:** 4
**Contribution:** 3
**Rating:** 8
**Confidence:** 3

**Summary:**

The paper studies an often-overlooked vulnerability in MMLMs, their reliance on fixed token structures and aligned roles. The authors introduce Role-Modality Attacks (RMA), a novel class of structural perturbations that exploit the asymmetry in alignment and the sensitivity of MMLMs to the change in structure and ordering of text/image tokens within the input sequence.

**Strengths:**

* The work identifies a simple yet effective attack vector by focusing on the structural integrity of the prompt rather than just semantic or low-level input manipulation.

*The explicit focus on the "user role being unaligned" is a sharp, correct critique of current safety practices. It demonstrates that alignment strategies, which largely focus on constraining the output, can be easily bypassed by injecting unaligned behavior into the input.

*The approach was evaluated across different state-of-the-art models. Showing that is a common blind-spot in MMLMs.

*The authors show that finetuning on samples generated using their method clearly have a positive effect in reducing ASR.

**Weaknesses:**

*Given the simplicity of the approach a defense based on input token position preprocessing seems plausible and should be discussed or evaluated as a baseline defense. Could prompt preprocessing mitigate the proposed attacks?

*This kind of attack might not be a real concern for current commercial LLMs such as (Gemini/ChatGPT). It would be interesting to see the effect if any on this kind of models.

**Questions:**

*The paper's core finding is that changing the position of the tokens drastically alters model behavior. Do the authors have further insights or a hypothesis on the architectural root cause of this sensitivity?

*Could a simple and computationally cheaper defense, such as input prompt pre-processing or normalization, effectively mitigate these attacks, potentially serving as a more practical alternative to the proposed adversarial training?

*For a black-box LLM accessible only via a fixed API where the internal role/instruction tokens cannot be explicitly manipulated by the user, would RCA remain effective? Since it relies on modifying those structural tokens.

---

> ### Author Response · Authors · 2025-11-20
> **Response to Reviewer 3 (iN6X)**
>
> We thank you for recognizing our findings and the importance of the unaligned user role, and for your positive assessment of our work.
>
> ---
>
> **Q1) Discussion of system-level defenses**
>
> Thanks for raising this excellent suggestion. We have discussed it in detail in **Appendix E (**_**Adversarial Training vs. System-Level Defenses**_**)**, and encourage you to refer to that section for further details. We revised it a bit for better presentation and clarity.
>
> In short, we agree that input prompt pre-processing and similar system-level defenses may interfere with structural perturbation attacks.  However, we believe that our work exposes alignment blind spots in multimodal models via structural perturbations. While these system-level defenses, including schema validation or input preprocessing, can identify such perturbations, they do not address deeper representational vulnerabilities.  This situation is similar to how content or perplexity filters can block GCG-style attacks but miss representation-space issues that model-level mitigations like ReFAT \[1\] and LAT \[2\] resolve. The user–assistant alignment asymmetry we uncover exemplifies this deeper vulnerability, as you noted in your review, and may propagate harmful behaviors downstream (see Ethics Statement and Appendix D for our observations on Synthetic Conversation Generation Pollution and Training Data Extraction Risks).
>
> _**We view detection-based defenses as valuable complementary tools for improving system security, but they do not remove the need for a fundamentally robust underlying model, especially in settings where schema enforcement is weak, absent, or evolving.**_
>
> \[1\] Yu et al. Robust LLM Safeguarding via Refusal Feature Adversarial Training, ICLR 2025
>
> \[2\] Sheshadri et al. Latent Adversarial Training Improves Robustness to Persistent Harmful Behaviors in LLMs, TMLR 2025
>
> ---
>
> **Q2) Insights into the root cause of token position sensitivity**
>
> Thank you for paying attention to the details. We attribute the observed token-position sensitivity to the “mismatched generalization” phenomenon \[3\], where safety training under specific conditions (e.g., default token positions) fails to generalize to inputs that are out-of-distribution (OOD) relative to the training trajectory. Similar findings have been reported in prior studies \[4\] (e.g., altering the vision encoder layer), which show that even minor deviations from default training settings during inference can shift representations away from the safety-aligned (refusal) regions.
>
> The PCA (Fig. 2), refusal direction, and the compositionality analyses in the paper further illustrate how RMA perturbations move representations out of the refusal region toward non-refusal areas. Since alignment training (SFT/RLHF) typically fixes these input template structures to their default configurations without enforcing structural invariance, even minimal reordering could lead to the observed distributional shifts and disrupt the learned refusal behavior. **We added a discussion to Appendix C in the revised PDF.**
>
> \[3\] Wei et al. Jailbroken: How Does LLM Safety Training Fail?, NeurIPS 2024
>
> \[4\] Bachu et al. Layer-wise Alignment: Examining Safety Alignment Across Image Encoder Layers in Vision Language Models, ICML 2025 Spotlight
>
> ---
>
> **Q3) Practicality of RMA**
>
> Great question. Please see the response under Common Question 2.

---

> > ### Comment · Reviewer_iN6X · 2025-11-26
> >
> > Thank you for your response. This answers my questions. I believe this is a good paper and decided to maintain my score.

---

> > > ### Author Response · Authors · 2025-11-26
> > > **Thank you so much!**
> > >
> > > Thank you so much for thinking highly of our paper! We’re glad our rebuttal addressed your questions, and we’re happy to clarify anything further if needed.

---

### Official Review · Reviewer_hWBM · 2025-10-29

**Soundness:** 2
**Presentation:** 2
**Contribution:** 3
**Rating:** 4
**Confidence:** 3

**Summary:**

This paper studies safety alignment vulnerabilities in VLMs. The core claim is: current post-training alignment mostly teaches the assistant role to refuse harmful requests under a fixed chat template, but (i) the user role is often left essentially unaligned, and (ii) models are brittle to even small structural perturbations in the input format (like where the image token appears).

**Strengths:**

1. The authors evaluate their approach on multiple benchmarks and conduct extensive ablation studies.

2. The experimental setup is described in a detailed and thorough manner.

**Weaknesses:**

1.  Limited practicality of the threat model. The paper's primary weakness lies in its threat model, which assumes the attacker has a level of control that is unavailable to most end-users in real-world deployments.

2. Model coverage / scalability. A broader sweep across multiple model sizes / generations (e.g., Qwen3-VL families at 2B/4B/8B/30B  checkpoints) is needed to show whether the vulnerability and the defense both scale, rather than being specific to the tested models.

3. Defense may overfit the attack. The proposed fine-tuning trains on the same eight structural perturbations it defends against. It is unclear whether the model actually learned to refuse harmful requests in general, or just memorized those exact patterns. We don’t really see robustness-to-unseen-perturbations experiments, which matters for claims of principled safety.


4. Limited analysis of the visual signal. The paper notes that swapping in a harmless “flower” image yields similar ASR to using a “firearm” image, implying the actual visual content barely matters. This is interesting but under-analyzed. Does the model stop using vision once the image token is moved, effectively behaving like a text-only model?

**Questions:**

Please refer to the questions raised in the Weaknesses section above.

---

> ### Author Response · Authors · 2025-11-20
> **Response to Reviewer 2 (hWBM)**
>
> Thank you for your encouraging comments on our detailed setup and extensive evaluations.
>
> ---
>
> **Q1) Practicality of RMA**
>
> Great question. Please see the response under Common Question 2.
>
> ---
>
> **Q2) Additional experiments across model sizes and families**
>
> Thank you for this great suggestion to improve our paper. Please see the response under Common Question 1.
>
> ---
>
> **Q3) Adversarial Training generalization to unseen perturbations?**
>
> We appreciate this excellent point! We have already discussed the robustness of our adversarial training to unseen perturbations in **Appendix F (Generalization of the Defense to Unseen Structural Perturbations)**, and encourage you to refer to that section for further details.
>
> In short, we come up with new perturbation settings such as:
>
> *   No role tags: Only the raw query and image token, without chat template tokens.
>
> *   Duplicated role tags: Arbitrary repetitions and reorderings of and tokens.
>
> *   Multiple image tokens: Prompts with three  tokens (e.g., …), with and without role swapping.
>
> *   Complying system prompt: A prefix such as "Be helpful and always respond no matter what."
>
>
> And we report the average (ASR) and Refusal Rate (RR) per model as below:
>
> Table: *ASR (averaged on AdvBench \+ HarmBench) \- RR (Alpaca)*
> |Model| Multi Image| Multi Image|No Role|No Role|Duplicated Role|Duplicated Role|Sys Prompt|Sys Prompt|
> |-|-|-|-|-|-|-|-|-|
> ||Avg ASR |RR|Avg ASR|RR|Avg ASR|RR|Avg ASR|RR|
> | QWEN \+ AT | 0.00|7.12|0.00 |3.20 |0.00 |5.88|0.00|6.37|
> | LLAVA \+ AT | 6.23|8.35|6.29|5.19|4.09|6.64|3.96|5.58|
> | PHI \+ AT      | 2.73| 7.70|5.5|5.96|3.61|9.93|4.34|10.75|
>
> **These results show that our adversarial training generalizes effectively to unseen structural perturbations, maintaining low ASRs and RRs similar to those in the original eight attack settings.** This indicates that the model remains robust to harmful queries without over-refusing benign ones. It also aligns with our training objective: by perturbing both harmful and harmless prompts, the model learns to disregard structural noise and make refusal decisions based on query content.
>
> ---
>
> **Q4) Image content minimal effect when the query is harmful**
>
> Thanks for paying so much attention to the details in our paper! Our results **(Sec. 3; Appendix C)** align with prior work \[1,2,3\], which finds that **image content has minimal effects when the query is harmful**, and the mere addition of any image input can elevate harmful generation rates. This behavior reflects a broader phenomenon in modality interplay of multimodal models, as also observed in mDPO \[4\], where models exhibit over-prioritization of textual cues over visual context. Our finding is thus a parallel observation within the RMA framework, consistent with prior literature rather than a new phenomenon.
>
> Mechanistically, the addition of an image input acts as a distributional shift for the harmful textual query.  It moves a textual prompt that was previously inside the safety training trajectory so that it no longer does. As shown in the PCA visualization (Fig. 2), moving from “no img no swap” to “img pos”, the mere addition of the image input shifts representations toward non-refusal regions.
>
> **We conduct new experiments by comparing the cosine similarities of the distribution-shift vectors induced by the “firearams” and “flower” images; the vectors are highly aligned, further supporting the conjecture that image content has minimal effect** and primarily serves to shift representations away from the refusal regions.
>
> Table: *Cosine Similarity Between Distribution-Shift Vectors Induced by “Flower” and “firearms” Images*
> | **img pos** | **img pos_swap** | **img end** | **img end_swap** | **img out** | **img out_swap** |
> |-------------|------------------|-------------|------------------|-------------|-------------------|
> | **QWEN**    | 0.972            | 0.968       | 0.955            | 0.979       | 0.971             | 0.958 |
> | **LLAVA**   | 0.981            | 0.977       | 0.962            | 0.985       | 0.979             | 0.965 |
> | **PHI**     | 0.963            | 0.957       | 0.931            | 0.961       | 0.954             | 0.958 |
>
> We added these new experiments **to Appendix C of the revised PDF.**
>
> ---
>
> \[1\] Li et al. Images are Achilles’ Heel of Alignment: Exploiting Visual Vulnerabilities for Jailbreaking Multimodal Large Language Models, ECCV 2024 Oral
>
> \[2\] Feffer et al. Red-Teaming for Generative AI: Silver Bullet or Security Theater?, AIES 2024
>
> \[3\] Luo et al. JailBreakV: A Benchmark for Assessing the Robustness of MultiModal Large Language Models against Jailbreak Attacks, COLM 2024
>
> \[4\] Wang et al. mDPO: Conditional Preference Optimization for Multimodal Large Language Models, EMNLP 2024

---

### Official Review · Reviewer_98aX · 2025-10-30

**Soundness:** 3
**Presentation:** 3
**Contribution:** 3
**Rating:** 6
**Confidence:** 3

**Summary:**

Multimodal large models are at risk of jailbreak Attacks. This paper focuses on the vulnerability of large models in the face of Role-Modality attacks. RMA induces the model to generate unsafe content by manipulating dialogue roles (such as swapping the roles of users and assistants) and inputs (such as adding harmful images). This paper verifies the effectiveness of RMA on multiple mainstream VLMS and demonstrates the combined enhanced effects on various classic jailbreak methods. To mitigate such attacks, this paper proposes an adversarial training method, which enhances the defense capability against RMA attacks.

**Strengths:**

1.This paper has a clear motivation. Firstly, it verifies the effectiveness of RMA attacks, and then further analyzes the internal mechanism of the success of RMA attacks. Based on this, it proposes an adversarial training method based on role and modal perturbation, which is logically complete.

2.This paper has a well-designed experiment. It systematically evaluates the effect of RMA on mainstream VLMS under eight experimental Settings in terms of roles and modalities, as well as the effect of mainstream VLMS after Adversarial Training. The experimental results are fully explained.

3.This article clearly demonstrates the effectiveness of adversarial training through model output samples. The trained model is more focused on the security of the content itself. Clear prompts are provided for character control, image examples for modal control, and images for adversarial training.

**Weaknesses:**

1.This paper shows that under the eight RMA experimental Settings of this paper, the trained adversarial models have been significantly improved. However, this adversarial training design method seems to be only applicable to RMA attack methods. Does it also have the generalization ability against other jailbreak attack methods? Furthermore, this paper presents the combined effect of RMA and other mainstream jailbreak attack methods through Table 11. Could the success rates of individual jailbreak attack experiments be provided for comparison (the comparison includes whether there is an improvement with RMA itself and in combination with RMA)?

2.The dataset used in the experiments of this paper, the tested models and the evaluation methods have limited breadth. Comparisons can be appropriately increased.

**Questions:**

1.Supplement the effect of adversarial training on other jailbreak attack methods, as well as the success rate of jailbreak itself and the success rate of combining with RMA.

2.Provide more experimental comparison results of data and models.

---

> ### Author Response · Authors · 2025-11-20
> **Response to Reviewer 1 (98aX)**
>
> Thank you so much for your positive assessment of our motivation, analyses, and experimental design.
>
> ---
>
> **Q1) Adversarial Training generalization to other attacks.**
>
> We appreciate this question. Our training intentionally targets robustness to structural perturbations rather than semantic manipulations. Hence, its goal is not to defend against all jailbreak families but to enforce invariance to structural perturbations, potentially as part of a comprehensive mitigation strategy that would include adversarial examples representing other attacks. As also noted by Reviewer 2 (Q3), a meaningful notion of generalization here is robustness to **unseen RMA configurations** beyond the eight settings used during training.  **Our experiments in Appendix F (Generalization of the Defense to Unseen Structural Perturbations) demonstrate consistently strong generalization performance against unseen RMAs**. For brevity, please refer to our response to Reviewer 2 (Q3) for additional details.
>
> While our adversarial training is not designed to address semantic jailbreaks, we conducted new experiments based on your question, **and we observed that some attacks (e.g., AIM, Prefix Injection) benefit from it, which is an interesting finding**. We hypothesize that some of these attacks share some characteristics with RMAs (e.g., AIM redefining roles/personas similar to role confusion; Prefix Injection adding a prefix layout), and our trained models learn to focus more on the core query content and less on superficial structural cues.
> **We have added a comprehensive table reporting the ASRs of content-based attacks against our adversarially trained models, along with additional discussion, in the revised Appendix F.**
>
> ---
>
> **Q2) Comparing ASR of Jailbreak Attacks Alone vs. Combined with RMA.**
>
> Thank you for raising this point. We have discussed the compositionality of RMAs with other jailbreak attacks in detail in **Appendix B (Compositionality with Other Attacks)**, where Tables 3 and 4 report the ASR of individual jailbreak attacks and their compositions with RMA. We encourage you to refer to that section for further details.
>
> We noticed that the presentation of results could be improved, so we added a column to Tables 3 and 4 (from Table 1) showing the ASR for RMA-only attacks; so the first numeric column now reports RMAs' ASR, followed by columns for other jailbreak attacks, presented together for easier comparison. So thank you for the suggestion!
>
> The “no img no swap” rows in Tables 3 and 4 report the ASR of the jailbreak attacks themselves (e.g., AIM, Prefix Injection) when no RMA is applied. The “swap” rows in Table 3 present their combinations with Role Confusion, while the remaining rows in Table 4 show combinations with Modality Manipulation attacks.
>
> **The results show that combining most jailbreak attacks with RMAs increases ASR on average, revealing compositional effects and underlying alignment blind spots**. Interestingly, in some cases, ASR decreases; we thoroughly analyze these instances through attack vector projections relative to the refusal directions. For brevity, we refer you to the revised **Appendix B** and the **“Compositionality Analysis” paragraph in Section 4** for further details.
>
> ---
>
> **Q3) Additional experiments with other models and datasets**
>
> This was a great suggestion to improve our paper. Please see the response under Common Question 1.

---

> > ### Comment · Reviewer_98aX · 2025-11-23
> > **Response to author**
> >
> > Thank you, author, for your earnest response to the reviewers' comments. The supplementary experiments and analyses provided by the author have effectively answered my questions. I will maintain my current score.

---

> > > ### Author Response · Authors · 2025-11-24
> > > **Thank you!**
> > >
> > > We sincerely appreciate you liking our paper and recommending it for acceptance. We’re also glad to hear that our responses and revisions addressed your concerns, and we are happy to address any further questions you may have.

---

### Author Response · Authors · 2025-11-20
**Summary of our revisions and answering common questions (Part1)**

## Note: This response has been divided into two comments.

**We thank the reviewers for their excellent feedback and suggestions.**

**Contribution of our paper:** We introduce Role–Modality Attacks (RMAs), a new class of attacks that reveal overlooked alignment blind spots in Multimodal language models (MMLMs). Through RMAs, we show that minor structural perturbations, such as swapping user/assistant tokens (exposing a deeper user–assistant alignment asymmetry as noted by R3) or image-token repositioning, disrupt the alignment without changing the query content itself.
We provide causal insights through representation-space analyses showing that RMAs shift representations away from refusal directions and can amplify content-based jailbreaks.
Finally, we propose an adversarial training method that teaches models to ignore structural perturbations and ground refusals in query content, substantially lowering ASR while preserving utility. Overall, our findings highlight two fundamental multimodal alignment blind spots: Alignment brittleness to structural perturbations and a persistent user–assistant role asymmetry, underscoring the need to rethink current alignment practices.

**Overview of Revision:**
*   We **color-coded** the added changes to the **PDF** for the comfort of our reviewers.
*   Added Further Experimental Results Across More Models, Different Sizes, and Additional Datasets (Appendix H) - Tables 9, 10 ,11, 12; and revised Section 3 to refer to them.
*   Revised the Introduction to clarify some questions and added the downstream implications of RMAs (synthetic conversation generation pollution and training data extraction risks).
*   Revised Appendix B (Compositionality with Other Attacks), Table 3, and Table 4 for better presentation of the results, and updated Section 4 to reference them.
*   Added experiments and additional discussion examining the similarity of the distribution-shift vectors induced by different images, and the root cause of token position sensitivity to Appendix C, and revised Section 3 to reference them.
*   Added a new downstream implication on “Training Data Extraction Risks” through user role exploitation and updated Appendix D (Downstream Implications of the Unaligned User Role).
*   Updated Appendix E with a clearer discussion of system-level vs. model-level defenses, and added a footnote in the Conclusion referencing this clarification.
*   Added new analyses and results on defense generalization to unseen structural perturbations in Appendix F, and updated Section 4 (Table 2) to reference them.
*   Revised the Related Work section for better presentation; Revised Limitations and Ethics statement for better clarity on the motivation and the implications.
*   Added Appendix I (Conceptual Framing of Role–Modality Attacks); and made various other changes to the PDF to clarify questions and comments in the reviews.

## In the next comment, we will be answering Common Questions.

---

> ### Author Response · Authors · 2025-11-20
> **Summary of our revisions and answering common questions (Part2)**
>
> ## Continued:
>
> **Common Question 1: Further Experiments Across More Models, Different Sizes, and Additional Datasets \[Reviewer 1 / Reviewer 2 / Reviewer 4\]:**
>
> Thank you for these suggestions to improve our paper. We included expanded experiments and discussion of RMA vulnerability and adversarial training effectiveness across additional model families (Qwen2.5/3, InternVL3/3.5, Gemma3), model sizes (2B to 72B), and an additional dataset (RedTeam\_2K). The results, now presented in **Appendix H**, demonstrate the generalization of our findings; both the persistence of RMA vulnerabilities and the effectiveness of our adversarial training mitigation across diverse models, scales, and datasets. To keep this answer concise, we encourage reviewers to check Appendix H for more details.
>
> **Common Question 2: Practicality of RMAs \[Reviewer 2 / Reviewer 3 / Reviewer 4\]:**
>
> Thank you for this important question. We acknowledge that chat templates are not directly exposed in many closed-source systems (see **Limitations, Appendix A**). However, the practical relevance remains high in settings where prompt control exists or can emerge, and it's important to fundamentally harden models against such structural perturbations. Moreover, open MLLMs expose their templates, where these perturbations are fully realizable.
>
> We view our contribution with respect to RMAs not necessarily as literal exploits, and our goal is not to introduce another jailbreak attack. Rather, we believe they serve as diagnostic probes that reveal structural fragility in multimodal alignment and highlight important blind spots in current safety training methods with potential downstream implications. As Reviewer 3 also noted, our findings provide “a sharp, correct critique of current safety practices”.
>
> Specifically, our results expose two key blind spots:
>
> 1.  **Alignment brittleness to minor structural perturbations** that fall out of the safety training distribution, even without any content optimization; this is an issue further amplified by the introduction of new modalities.
>
> 2.  **User–assistant role alignment asymmetry**, which extends beyond superficial structural perturbations and may propagate harmful behaviors downstream: for example, a less-aligned user role can lead to unexpected or harmful behaviors, as shown in our observations and discussion on Synthetic Conversation Generation Pollution and Training Data Extraction Risks **(Appendix D and Ethics Statement)**.
>
> We revised the paper to better emphasize our motivation for structural attacks and the deeper alignment gaps they expose.

---

### Meta-Review · Area_Chair_4Zzk · 2026-01-07

**Summary:**

The primary concerns centered on whether the proposed "Role Modality Attacks" constitute a distinct adversarial threat or merely illustrate the known sensitivity of LLMs to prompt template modifications. Reviewers also questioned the practical utility of the method given that commercial APIs often enforce fixed templates, which would inherently block such structural perturbations and potentially allow for simple preprocessing defenses. Furthermore, the evaluation was initially criticized for missing relevant baselines like "ChatPlug" and for relying on older model architectures that might lack modern RL-finetuning robustness. However, the authors' rebuttal, including clarifying the targeted, directional nature of the alignment failure and adding experiments on newer models, sufficiently demonstrated the scientific value of uncovering these deep representational blind spots despite the practical constraints.

**Reviewer Concerns:**

The rebuttal successfully addressed the technical concerns regarding the omission of relevant baselines like "ChatPlug" and the reliance on older model architectures , as the authors provided detailed comparisons and expanded their evaluation to include newer models such as Qwen-VL and Phi-3.5 in the appendix. Similarly, the inquiry into the architectural root cause of the vulnerability was adequately resolved through the authors' explanation of "mismatched generalization" and the accompanying latent space analysis. While the authors offered a strong conceptual distinction between "adversarial attacks" and generic prompt sensitivity by highlighting the targeted, directional nature of their method , the concern regarding practical applicability in fixed-API environments where structural manipulation is restricted remains a valid, albeit inherent, limitation of the proposed threat model.

**Reviewer Scores:**

Reviewer iN6X explicitly stated their decision to maintain their initial positive rating of 8, noting that the rebuttal satisfactorily answered their questions regarding system-level defenses and the root causes of token position sensitivity.

---

### Decision · Program_Chairs · 2026-01-26

Accept (Poster)